# Botulinum Neurotoxin A Injections in Spasmodic Entropion: A Clinical Retrospective Cohort Study

**DOI:** 10.3390/toxins17080383

**Published:** 2025-07-31

**Authors:** Brigitte Girard, Fabienne Carré, Simon Begnaud

**Affiliations:** 1Ophthalmology Department, Tenon Hospital, AP-HP, 4 rue de la Chine, 75020 Paris, France; simon.begnaud@gmail.com; 2Health Faculty Pitié-Salpétrière Site, Bd de l’Hôpital, Sorbonne University, 91-105, 75013 Paris, France; 3Hôpital Privé Armand Brillard, 3 Avenue Watteau, 94130 Nogent sur Marne, France; fabienne.carre@aphp.fr; 4ENT Department, Pitié-Salpêtrière Hospital, AP-HP, 46-83 Boulevard de L’hôpital, 75013 Paris, France; 5Ophthalmology Department, Ophthalmology National Hospital Center, 28 rue de Charenton, 75012 Paris, France; 6Fellow Fondation Bettencourt Schueller, 92200 Neuilly-sur-Seine, France

**Keywords:** botulinum neurotoxin, entropion, spasmodic palpebral entropion, eyelids, blepharospasm, noninvasive treatment

## Abstract

While surgical procedure has been considered as the golden standard treatment for spasmodic entropion, Botulinum Neurotoxin A can be indicated in the treatment of spasmodic entropion for fragile elderly patients. This retrospective cohort study included 50 outdoor patients treated for spasmodic entropion, for whom palpebral surgery was recused. The intent of the present study was to describe an alternative outdoor treatment, to detail precisely the Botulinum Neurotoxin (BoNT) treatment pattern, the dosage of BoNT needed, the frequency of re-injection, the efficiency and the complications encountered. Fifty patients, 87.9 years old in average (±14.3) have been injected with BoNT. The average total dosage of BoNT is 7.62 ± 1.38 units of Incobotulinum, 10.2 ± 1.03 units of Onabotulinum and 17.2 ± 1.33 Speywood-units of Abobotulinum. Spasmodic entropion resolved in 3 ± 2 days after the BT injection. The average for re-injection is every 4.25 ± 1.30 months. By adjusting age and total dose, we have not been able to show any statistically significant relationship between time needed for re-injection and type of botulinum toxin A (*p* = 0.59). Patients with spasmodic entropion have responded significantly to BoNT injection. No systemic complications have been reported in this study. BoNT treatment is safe and effective for fragile elderly patients with spasmodic entropion and can be proposed instead of surgery or while waiting for their procedure.

## 1. Introduction

Spasmodic entropion is characterized by inward turning of the lower eyelid secondary to an ectopic contraction of the inferior Orbicularis Oculi Muscles (OOM). Entropion can be isolated, and classified as congenital, involutional, senile, cicatricial or spasmodic [1]. It may occur secondary to severe Essential Blepharospasm (EBS). EBS is a focal cerebral dystonia characterized by unintentional and bilateral eyelid closure induced by contractions of OOM [2,3]. Spasmodic entropion will complicate EBS management because of ocular complications such as eyelashes rubbing, keratitis and corneal ulcerations [4]. Surinfection can be responsible for conjunctivitis or corneal abscess. Corneal pain and photophobia are part of the process of eyelid reflex closure, increasing palpebral spasm, worsening symptoms of EBS. Spasmodic entropion has to be treated to reduce ocular complications, and to improve quality of life.

Senile entropion is bound by involutional lower eyelid changes such as lower eyelid retractor dehiscence, canthal tendon laxity, pre-septal OOM moving over pre-tarsal OOM, aged-related enophthalmos and tarsal plate degeneration [4,5]. Treatment of senile entropion usually consists of surgical correction of the horizontal laxity with canthal tendon shortening or correction of vertical laxity with lower lid retractor reinsertion [4,6,7,8,9,10]. The usual treatment consensus for palpebral entropion is surgical with satisfactory outcomes. In some cases, surgery is contra-indicated and a better understanding of entropion etiology and physiopathology allow different treatment of spasmodic entropion, as blepharospasm is an abnormal intermittent movement. For forty years, Botulinum Neurotoxin type-A (BoNT/A) injections have been used to release muscle contractions induced by cerebral dystonia or other movement disorders [11,12,13]. BoNT/A inhibits acetylcholine release at the skeletal neuromuscular junction and induces a transient flaccid paralysis [14]. BoNT/A also inhibits ectopic muscle contractions at the neuromuscular junction. Over the last forty years, BoNT/A injections have become increasingly used in the treatment of facial dystonia, blepharospasm, and other movement disorders [11,15,16]. Significant response rates and the satisfying safety profile of BoNT/A injections are well established, with transient and easily overcome complications [16,17,18,19]. As an alternative to surgical treatment of spasmodic entropion, Carruthers first injected the spastic lower lid with BoNT/A with encouraging resolution of symptoms a few days after injections [20,21]. Later, these encouraging results were reproduced and safety was confirmed [21,22,23,24]. Iterative reinjection leads to a sustained effect on spasmodic entropion [19]. These studies are scattered in the literature and examine a limited number of patients in non-systematic protocols.

The aim of this present clinical study is to evaluate the efficiency and safety of BoNT/A injections as an alternative treatment for spasmodic entropion in elderly patients who have refused or have been declined for surgery.

## 2. Results

Fifty outdoor patients, ineligible for palpebral surgery considering their physical condition, received repeated and scheduled Botulinum Neuro-Toxin (BoNT) injections to treat spasmodic entropion. A total of 45 patients (90%) had unilateral senile spasmodic entropion (21 right, 24 left) while 5 patients had bilateral entropion. The mean age of patients was 87.9 ± 14.3 years, ranging from 66 to 102 (Table 1 and Table 2). The sex ratio was 0.43 (35 women to 15 men). Mean age at onset of symptoms was 78.7 ± 12.4 years. Average dosage of BoNT injected in the involved lower lid was 7.62 ± 1.38 units of Incobotulinum Toxin (Xeomin^®^) for 23 patients; 10.2 ± 1.03 units of Onabotulinum Toxin (Botox^®^) for 17 patients and 17.2 ± 1.33 Speywood-units of Abobotulinum Toxin (Dysport^®^) for 10 patients.

Spasmodic entropion resolved in 3 ± 2 days after BoNT/A injections for all patients. Average recurrence and re-injection time was every 17 ± 5.2 weeks ranging from 12 to 24 weeks. Time before reinjection was not statistically different for the three brands of BoNT/A after adjustment for age and total injected dose (*p* = 0.59) (Table 3).

No severe or systemic adverse event was reported during the study. The most frequent complication was an ecchymosis at the injection site for 10% of patients and was not more severe or more frequent for the patients treated with anticoagulants or aspirin. In one case, entropion had been overcorrected in ectropion, lasting for a month in a patient treated with Abobotulinum Toxin, and spasmodic entropion reappeared four months later. No diplopia or ptosis was reported.

## 3. Discussion

Spasmodic Entropion is an ectopic contraction of the inferior OOM responsible for inward tilt of the inferior palpebra. It can be isolated or associated with senile entropion.

Before proposing surgery or BoNT injections, it is necessary to evaluate the etiology of the entropion and the equilibrium of the forces present to choose the good therapeutic procedure. Inferior palpebra is physiologically molded against the eyeball and stretched by tarso-ligamentary strap. When the orbicular oculi muscle contracts to close the eyelids, the lower palpebral margin is raised up by sliding on the eyeball harmoniously. With age, three phenomena interfere with this balance: tarso-ligamentary strap becomes loose, inducing palpebral laxity, bone walls are reshaped with a blunting of the orbital bone edge and orbital fat is melting, inducing senile enophthalmia [25,26,27,28,29]. Inferior palpebral position results in the balance between horizontal, vertical and antero-posterior forces associated with the position of the eyeball. Horizontal forces result in the strength of internal and external canthus attachments. Assessment of the medial canthal tendon is obtained by pulling the inferior eyelid laterally on a horizontal plane outwardly, and measuring the migration of the lower tear meatus which should not be greater than 2 mm [30]. Assessment of the lateral canthal tendon is obtained by pulling the inferior eyelid on a horizontal plane inwardly, and the external canthus migration is observed. The test is pathological if deplacement of lateral canthus is over 2 mm. Vertical forces result in the integrity and thickness of the tarsus [31], attachment and trophycity of the Palpebral Retractors Muscles, and integrity of cutaneo-muscular ligaments. It should be appreciated by the persistence of inferior eyelid fold in downward gaze. However, in elderly patients, association of orbital fat melting and cachexia are responsible for senile enophtalmia and oculo-palpebral diastasis [27,28,29,30,31,32]. It can be measured by exophtalmometer or orbital CT scanner. These ageing modifications, constitutional orbital fat melting and orbital ridge bone reshaping, responsible for the sinking of the eyeball in the orbit, induce an inefficient contraction of the Pre-Septal Orbicularis Oculi Muscle which slides over the Pre-Tarsal OOM, modifying the normal movement of inferior palpebra. The lower eyelid, instead of sliding up against the surface of the eyeball, tilts inward inducing senile entropion of the lower eyelid [33,34,35]. In this anatomical situation, entropion is perpetuated with spasticity related to pain induced by the friction of the eyelashes on the eye surface (conjonctiva and cornea). Senile entropion and spasmodic entropion can be associated, although some authors consider spasmodic entropion dystonia its own variant of essential blepharospasm [36]. In this case, no ageing modifications of the tarsus, orbital rim, orbital fat and tarso-ligamentary strap are observed.

If surgery has appeared as the golden treatment for senile entropion [6,7,8,9,10,37,38], different surgeries have been proposed depending on the entropion etiologies, the quality of the tissues and the tensile forces presents [35,39,40]. In absence of consensual surgical guidelines for entropion treatment, because of recurrence for 10% to 20% of surgical patients, and because of extended surgical delays, BoNT injections has appeared as alternative therapy [41,42,43,44,45].

Our first retrospective observational study has suggested the efficiency and safety of BoNT/A injections in the lower lid for spasmodic entropion associated with EBS in eight patients [23]. After this first study, we extend our indications of BoNT injections in all spasmodic entropion isolated or related with EBS, associated or not to senile entropion with surgery contra-indication. Contrary to senile entropion, spasmodic entropion is not only due to passive laxity driving imbalance in the forces of palpebral stability but in local spasticity.

In this study, fifty patients affected by spasmodic entropion either associated to EBS or to senile entropion, declined for surgical procedure, were included for BoNT injection treatment after informed consent. Whereas life expectancy in France is 82 years, average age of onset of entropion in this study was 78.7 +/− 12.4 years, close to the French life expectancy age. For these old patients affected by poor physical conditions, and sometimes cachexia, surgery was not recommended. As expected, patients recruited in this study were much older than patients without contra-indication for surgery recruited for entropion surgery in the literature; 78 years old versus 70 years old according to different surgical studies [34,40,45].

As an alternative to surgery, we proposed BoNT/A injections to relieve at least the spasticity associated with entropion with scheduled repeated injections. While today, BoNT/A injections for spasmodic entropion remain a controversial treatment, entropion was one of the first BoNT injections indicated, with an excellent safety and efficiency profile [21,22,41,46].

In the literature, BoNT/A injection protocols delivered in the spasmodic lid are heterogeneous, concerning few patients. There is no consensus concerning the total dose delivered, nor the NTBo brand, nor the injection points. The three BoNT/A brands commercially available in Europe have been used. As it is well established, there is no correspondence in units between the three brands, as a result of manufacturing secrets. Usually, the units used for IncoBotulinum Toxin and Onabobotulinum Toxin are nearly similar in clinical practice; although units for Abobotulinum Toxin (called Speywood units) are considered with a coefficient three times higher [47,48]. The total dose delivered in the spasmodic lid varies from 10 to 20 units of Onabobotulinum toxin (Botox^®^) in one injection puncture of 0.1 mL or split in five micro injections of 0.02 mL [23]. The total dose delivered for Abobotulinum Toxin (Dysport^®^) varies from 20 to 50 Speywood units [19,22,23,43,49]. Few studies relate their experience about Incobotulinum Toxin (Xeomin^®^). We have presented the first cases treated by Incobotulinum Toxin in 1988 [23]. In our preliminary study, we used 10 units of Incobotulinum Toxin concentrated at 100 μ/mL, split into four injection punctures of 0.025 mL.

Considering our experience and these literature data, we standardized the protocol to propose a reproducible and simple technique. In this study, we injected 0.025 mL repeated in four injections sites, in inferior palpebra, 2 mm under the eyelash line, strictly subcutaneously, above OOM plane, with respect to the lacrimal meatus. This procedure realizes an BoNT/A injection in preseptal OOM, the muscle which overlap the pretarsus OOM. The total dose of Onabotulinum or Incobotulinum Toxin (Botox^®^ and Xeomin^®^, respectively) was 10 U in the lower palpebra, split into four sites of 2.5 U. The total dose injected of Abobotulinum toxin (Dysport^®^) was 20 Speywood units split into four spots of 5 Speywood units. Each brand was diluted with NaCl 0.9% to obtain concentration as follow: Onabotulinum Toxin or Incobotulinum Toxin were concentrated at 100 U/mL; Abobotulinum Toxin were concentrated at 200 Speywood Units/mL.

As reported in literature, BoNT/A injections correct entropion from no delay to 4 days post injection. The immediate result cannot be a BoNT/A effect, because the toxin has to be internalized in the terminal axon before blocking liberation of Acetylcholine in the neuromuscular synapse, but rather the mechanical everting action of volume injected in the anterior lamella.

This BoNT injection technique for entropion prevents overcorrection, ectropion of the lacrimal meatus, and diffusion towards the intra-orbital structures, especially the oculomotor muscle [23]. It is important to remember that the lower the volume which is injected at a precise location, the less BoNT/A will diffuse toward intraorbital muscles or Zygomaticus Muscles, pointing out the necessity of increasing BoNT concentration [43].

Among the three brands of BoNT/A commercialized and used, we found no statistical difference in term of efficacy with this dosage. The use of non-complexing Incobotulinum Toxin (Xeomin^®^) is preferred to avoid neutralizing antibodies (49). The choice of the brand was left to the physician practice, based on spasm intensity, and is a bias for the results. For severe spasms, Abobotulinum Toxin (Dysport^®^) was preferred. For minimal to moderate spasms prior to 2008, Onabobotulinum toxin (Botox^®^) and after 2008, Incobotulinum Toxin (Xeomin^®^) were preferred.

While we found a recurrence in 100% of cases at four to six months, Iozzo et al. found only 27% recurrence of spasmodic entropion after 6 months of BoNT/A injection [43]. The duration of effectiveness is increased after several injections leading to delaying the reinjection [19]. Anatomical analyses showed no disruption of muscle architecture and respect of the muscular fiber architecture after BoNT/A injections [22]. Recall bias is limited in our study as all patients showed up for the follow-up.

According to our study, BoNT/A injections were effective to relieve entropion after 3 days consistent with the literature ranging from 0 to 15 days [23,41,43]. Treatment effect lasts 17 weeks, ranging from 12 to 24 weeks in our study, which is comparable with other works [22,24,41,43].

The injection technique is simple and repeatable. Four injections of 0.025 mL are recommended, delivered 2 mm under eyelash line, subcutaneously, in the preseptal OOM, with respect to the lacrimal meatus. This technique prevents ectropion of the lacrimal point, orbital diffusion towards Oculomotors Muscles and overcorrection (Figure 1). It is important to remember that the lower the volume which is injected at a precise site, the less BoNT/A will spread toward intra-orbital muscles or cheeks, preventing facial drop or diplopia [22].

## 4. Conclusions

This study confirms that BoNT/A injections are an efficient and safe alternative to surgery for Spasmodic Entropion in elderly patients. This outdoor treatment can be proposed at the first consultation, especially for fragile patients ineligible for surgery, and can be repeatedly programmed. Prospective randomized trials or standardized protocols with prospective randomized trials or standardized protocols would corroborate and lend precision to these conclusions.

## 5. Materials and Methods

We led an observational retrospective study. Informed consent was obtained from all subjects involved in the study and the study was approved in accordance with the reference methodology of the National Commission for Data Protection and Liberties (CNIL-France, no. 20240219143112).

Patients referred for abnormal palpebral movement to the Ophthalmologic Department of the Hospital were included in this observational study from January 2002 to December 2021 after diagnosis of spasmodic entropion and refusal by the patient of a surgical treatment, after complete and medical information or with contraindication for surgery because of a poor physical condition (ECOG performance status 3 or 4), after anesthetist’s advice. Patients were informed of the objectives and complications of BoNT/A injections. A questionnaire was filled in with the practitioner to avoid any contraindication to BoNT/A injection (myasthenia or other neuro-muscular disease). Patients with no follow-up after injection were excluded. Anticoagulant treatment was not interrupted.

For each patient, we recorded sex, age, type of entropion, and beginning of spasmodic entropion symptoms. After clinical examination including palpebral laxity to evaluate entropion’s strength, Schirmer test, break-up time and slit lamp examination to evaluate keratitis, the therapeutic choice was explained to the patient between surgery or BoNT injection. With their consent, patients with a contraindication for surgery, or because of surgery refusal, were programmed for BoNT injections. BoNT/A re-injections were performed as necessary, by patient’s choice.

The date of first BoNT/A injection, BoNT/A brand and concentration used, the volume and total dose injected, the frequency of re-injections needed to guarantee an acceptable quality of life for the patient have been recorded. The efficiency and complications were reported after a monitoring phone call two weeks after injections.

BoNT/A dilution with NaCl 0.9% was performed by the practitioner to achieve the desired concentration. BoNT/A was diluted at 100 μ/mL for Incobotulinum Toxin (Xeomin^®^, Merz Pharmaceuticals, Frankfurt am Main, Germany) and Onabotulinum Toxin (Botox^®^, Abbvie, North Chicago, IL, USA) and at 200 Speywood μ/mL for Abobotulinum Toxin (Dysport^®^, Ipsen, Paris, France). Injections were performed with 30-gauge hypodermic needle and a 1 mL three-piece syringe. Total volume injected was 0.08 mL to 0.1 mL, split into four spots. No local anesthesia was required. If the injection used was painful, an ice cube was applied on the lower lid before injection. Ectopic inferior preseptal OOM, slid over pretarsal OOM, was slowly injected, bevel up, with 4 subcutaneous punctures of 0.020 to 0.025 mL just below the eyelid margin, 2 mm under eyelashes, avoiding the inferior lacrimal punctum (Figure 1). The choice of the BoNT/A brand was at practitioner’s discretion, depending on the strength of the spasm.

### Statistical Analysis

We carried out a linear regression, with the outcome variable “Time between each injection” and the explanatory variables “Type of botulinum toxin A, Age and Total dose”. The covariates “Type of botulinum toxin A”, “Age” and “Total dose” were defined a priori. The candidate covariates were included in a Least Absolute Shrinkage and Selection Operation (LASSO) penalized regression model.

## Figures and Tables

**Figure 1 toxins-17-00383-f001:**
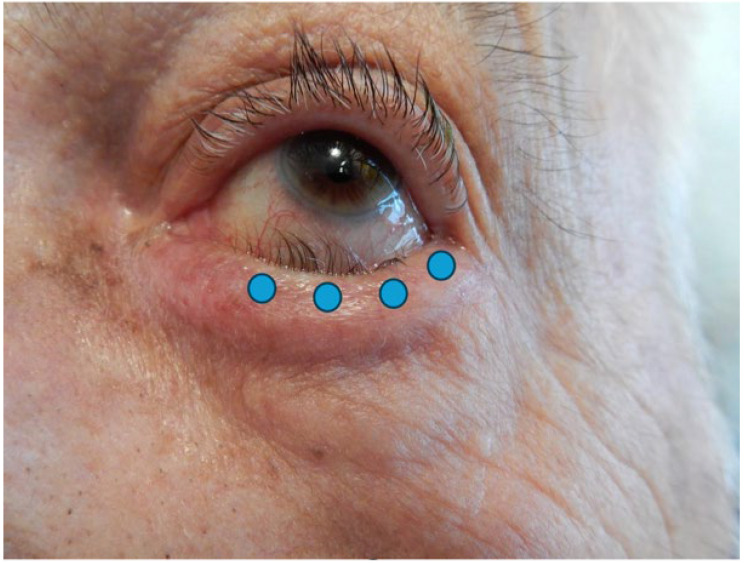
Photograph of an entropion of the left lower eyelid. Blue dots represent injection points.

**Table 1 toxins-17-00383-t001:** Patients’ characteristics (n = 50). Quantitative variables.

	Mean (sd)	Median [Q25-75]	Min	Max
Age at BONT/A injections (years)	87.90 (14.30)	90.50 [83.00; 98.00]	66.00	102.00
Age at onset of entropion (years)	78.70 (12.40)	80.00 [67.00–88.00]	52.00	101.00
Time between each injections (weeks)	17.00 (5.2)	16 [12.00; 24.00]	5.00	24.00

**Table 2 toxins-17-00383-t002:** Patients’ characteristics (n = 50). Qualitative variables.

Gender	Female	35 (70%)
	Male	15 (30%)
Entropion side	Left	24 (48%)
	Right	21 (42%)
	Bilateral	5 (10%)
Type of botulinum toxin A	Incobotulinum	23 (46%)
	Onabotulinum	17 (34%)
	Abobotulinum	10 (20%)

**Table 3 toxins-17-00383-t003:** Delay between two injections depending on each botulinum toxin molecule.

BoNT/A	X	B	D	*p*-Value
N (patients)	23	17	10	
Time between injections in weeks (sd)	18 (±6)	18 (±5)	16 (±4)	0.59

X: Incobotulinum toxin (Xeomin®); B: Onabotulinum toxin (Botox®); D: Abobotulinum toxin (Dysport ®); sd: standard deviation.

## Data Availability

The original contributions presented in this study are included in this article. Further inquiries can be directed to the corresponding author.

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
