# Peer review of "Botulinum Neurotoxin A Injections in Spasmodic Entropion: A Clinical Retrospective Cohort Study"

_toxins, 2025, doi:10.3390/toxins17080383_

Round 1

Reviewer 1 Report

Comments and Suggestions for Authors

Although using botulinum toxin A to treat spasmodic entropion have many publications to prove its efficacy in clinical practice. This manuscript of “Botulinum Neurotoxin A Injections In Spasmodic Entropion: A

Clinical Retrospective Cohort Study” do add useful data about different brands of toxin and average duration of reinjection (roughly equal to duration of toxin effect).

The mean age of onset of entropion is 78.7 year old and age of botulinum toxin A injection to be 87.9 year old in this study , further point out the value of using botulinum toxin A as alternative for old age patient group for entropion treatment with minimal downtime, side effects and good efficacy. 

In figure 1, please use dots or cross to indicate the four injection points for easy understanding.

Reviewer 2 Report

Comments and Suggestions for Authors

1.    The scientific writing should be in the passive voice.
    2.    The keywords should focus more on the core aspects of the study.
    3.    The basis on which the patient sample size was determined should be clarified.
    4.    The prognosis of the study should be reported.
    5.    Table 1 and 2 have no title.
    6.    In the Discussion section, there should be a correlation between the prognosis of surgical treatment and botulinum injection.
    7.    Figure 1 has no title.
    8.    The inclusion and exclusion criteria should be clearly stated.
    9.    Ethical approval should be reported.
    10.    The Conclusion and Recommendations sections are missing.
    11.    References need to be updated.

Reviewer 3 Report

Comments and Suggestions for Authors

The manuscript presents a retrospective cohort study evaluating the efficacy and safety of Botulinum Neurotoxin A (BoNT/A) injections as an alternative treatment for spasmodic entropion in elderly patients who are ineligible for surgery. The study is well-structured and addresses an important clinical gap, particularly for fragile elderly patients. This study provides valuable insights into the use of BoNT/A for spasmodic entropion in a challenging patient population. With the suggested revisions, the manuscript will be more robust and impactful for clinicians and researchers. I commend the authors for their work and look forward to seeing the revised version.

In abstract section, the statistical significance of the results (e.g., p-values) should be explicitly mentioned in the abstract to strengthen the conclusions.

In Introduction section, a clearer distinction between spasmodic and senile entropion should be stated, as the two conditions are often conflated in the text.

A brief mention of the limitations of existing BoNT/A studies (e.g., small sample sizes, lack of standardized protocols) would better justify the need for this study.

In Study design, the potential biases like selection bias, recall bias should be discussed.

In patient selection, the criteria for "fragile elderly patients" should be more explicitly defined (e.g., specific comorbidities, functional status).

In injection protocol, the rationale for choosing 4 injection sites (as opposed to fewer or more) could be elaborated.

In statistical analysis, the lack of a control group (e.g., patients treated with surgery or placebo) limits the strength of the conclusions.

In demographics, the sex ratio (0.43, 35 women to 15 men) raises questions about potential gender-based differences in treatment response. This could be explored further.

The resolution of entropion within 3 days is impressive, but the claim that this is due to the "mechanical exerting action of volume injected" is speculative. Supporting references or a discussion of possible mechanisms would strengthen this point.

In re-injection interval,  the lack of statistical significance between toxin types (p = 0.46) should be interpreted cautiously given the small sample sizes for each subgroup (e.g., only 10 patients received Abobotulinum).

The reported complications are minimal, which is reassuring. However, the absence of systemic complications should be contextualized with the study's limited power to detect rare adverse events.

Suggestions for prospective randomized trials or standardized protocols would strengthen the conclusion.

Table 2 could include confidence intervals for the re-injection intervals to provide a better sense of variability.

There are occasional grammatical errors and awkward phrasings (e.g., "recused for surgery" should likely be "declined surgery" or "were ineligible for surgery"). A thorough proofreading would improve readability.

Round 2

Reviewer 2 Report

Comments and Suggestions for Authors

Thanks for doing modification 

Reviewer 3 Report

Comments and Suggestions for Authors

No further comments